# bigPint: A Bioconductor visualization package that makes big data pint-sized

Lindsay Rutter [1]*, Dianne Cook [2]

**1** Bioinformatics and Computational Biology Program, Iowa State University, Ames, Iowa, United States of America, **2** Econometrics and Business Statistics, Monash University, Clayton, VIC, Australia

* lindsayannerutter@gmail.com

## Abstract

Interactive data visualization is imperative in the biological sciences. The development of independent layers of interactivity has been in pursuit in the visualization community. We developed bigPint, a data visualization package available on Bioconductor under the GPL-3 license (https://bioconductor.org/packages/release/bioc/html/bigPint.html). Our software introduces new visualization technology that enables independent layers of interactivity using Plotly in R, which aids in the exploration of large biological datasets. The bigPint package presents modernized versions of scatterplot matrices, volcano plots, and litre plots through the implementation of layered interactivity. These graphics have detected normalization issues, differential expression designation problems, and common analysis errors in public RNA-sequencing datasets. Researchers can apply bigPint graphics to their data by following recommended pipelines written in reproducible code in the user manual. In this paper, we explain how we achieved the independent layers of interactivity that are behind bigPint graphics. Pseudocode and source code are provided. Computational scientists can leverage our open-source code to expand upon our layered interactive technology and/or apply it in new ways toward other computational biology tasks.

## Author summary

Biological disciplines face the challenge of increasingly large and complex data. One necessary approach toward eliciting information is data visualization. Newer visualization tools incorporate interactive capabilities that allow scientists to extract information more efficiently than static counterparts. In this paper, we introduce technology that allows multiple independent layers of interactive visualization written in open-source code. This technology can be repurposed across various biological problems. Here, we apply this technology to RNA-sequencing data, a popular next-generation sequencing approach that provides snapshots of RNA quantity in biological samples at given moments in time. It can be used to investigate cellular differences between health and disease, cellular changes in response to external stimuli, and additional biological inquiries. RNA-sequencing data is large, noisy, and biased. It requires sophisticated normalization. The most popular open-source RNA-sequencing data analysis software focuses on models, with little emphasis on integrating effective visualization tools. This is despite sound evidence that RNA-sequencing

**Data Availability Statement:** The four public datasets used in our recent methods paper are available online: Three are deposited on the NCBI Sequence Read Archive with accession numbers SRA000299, PRJNA318409, and SRA048710. One

is deposited on the NCBI Gene Expression Omnibus with accession number GSE61857. The original dataset we used in our recent methods paper is also available on the NCBI Gene Expression Omnibus with accession number GSE121885. The bigPint package itself comes with two of the aforementioned RNA-sequencing datasets as examples. The package can be downloaded from the Bioconductor website (https://bioconductor.org/packages/devel/bioc/html/bigPint.html). As linked in the Bioconductor website, the bigPint package also has a vignette website (https://lindsayrutter.github.io/bigPint), where users can follow reproducible code to install the software and use the example datasets to create bigPint graphics and follow an example analysis pipeline.

**Funding:** The authors received no specific funding for this work.

**Competing interests:** The authors have declared that no competing interests exist.

data is most effectively explored using graphical and numerical approaches in a complementary fashion. The software we introduce can make it easier for researchers to use models and visuals in an integrated fashion during RNA-sequencing data analysis.

This is a *PLOS Computational Biology* Software paper.

## Introduction

Interactive data visualization is increasingly imperative in the biological sciences [1]. When performing RNA-seq studies, researchers wish to determine which genes are differentially expressed between treatment groups. Interactive visualization can help them assess differentially expressed gene (DEG) calls before performing any subsequent functional enrichment analyses. New visualization tools for genomic data have incorporated interactive capabilities, and some believe this trend could enhance the exploration of genomic data in the future [2]. Despite the growing appreciation of the inherent value of interactive graphics, the availability of effective and easy-to-use interactive visualization tools for RNA-seq data remains limited.

Interactive visualization tools for genomic data can have restricted access when only available on certain operating systems and/or when requiring payment [3–5]. These limitations can be removed when tools are published on open-source repositories. Indeed, the Bioconductor project aims to foster interdisciplinary scientific research by promoting transparency and reproducibility while allowing software content to be used on Windows, MacOS, and Linux [6]. Bioconductor software is written in the R programming language, which also provides statistical and visualization methods that can facilitate the development of robust graphical tools [7]. Several interactive visualization methods for genomic data have been developed using Shiny, which is also based on the R programming language [8–10].

We recently developed bigPint, an interactive RNA-sequencing data visualization software package available on Bioconductor. In the current paper, we will now explain the technical innovations and merits of the bigPint package, including new interactive visualization techniques that we believe can be helpful in the development and usage of future biological visualization software.

### Design and implementation

**Quick start.**   For users who would like to immediately try out the package hands-on and apply bigPint graphics to their data, we recommend consulting the example pipeline (https://lindsayrutter.github.io/bigPint/articles/pipeline). This pipeline uses reproducible code and sample data from the bigPint package, so you can smoothly follow along each line of example code. For additional details, we recommend users to view articles in the Get Started tab on the package website (https://lindsayrutter.github.io/bigPint).

### Basic input

Users can choose whether to input their data in *SummarizedExperiment* format using a single parameter *dataSE* or input their data as a combination of a parameter *data* and *dataMetrics*. In general, the *data* format corresponds to the *assay(SummarizedExperiment)* format and contains the read counts for all genes of interest. The value in row *i* and column *j* should indicate how many reads have been assigned to gene *i* in sample *j*. This is the same input format

required in popular RNA-seq count-based statistical packages, such as DESeq2, edgeR, limma, EBSeq, and BaySeq [11–15].

In general, the *dataMetrics* object corresponds to the *rowData(SummarizedExperiment)* format and should be a subset of the data (usually DEGs) where each case includes quantitative values of interest (such as fold change and FDR). This information can be easily derived from popular RNA-seq numerical analysis packages. Again, this framework allows users to work smoothly between visualizations in the bigPint package and models in other Bioconductor packages, complying with the belief that the most efficient way to analyze large datasets is to iterate between models and visualizations.

## Original features

**Independent layers of interactivity.** The Bioconductor community advanced the boundaries of biological visualization in the past and generally believes that modern interactive technology must be incorporated to continue these advancements [6]. We will define the term *geom-drawing interactivity* to indicate user queries that draw geoms (graphical representations of the data, such as lines, hexagons, and points). This could mean the user adjusts sliders or selects buttons to draw a subset of the data from the database as geoms (such as points). We will define the term *geom-manipulating interactivity* to indicate user queries that alter already-drawn geoms. This could mean the user hovers over a geom (such as a hexagon) and obtains its associated metadata (such as the names of its contained genes). It could also mean the user zooms and pans to further alter how already-drawn geoms are displayed.

Our package introduces what we believe is a fairly new interactive visualization technology that is useful in the exploration of large biological datasets. Our technique allows for two independent layers of interactivity, for the foreground and background of the plot respectively. Each layer can include both *geom-drawing* and *geom-manipulating* interactivity. Our new

**Table 1. Resources for users about bigPint interactive graphics.**

| Plot | Figure | Video | Application |
|------|--------|-------|-------------|
| Scatterplot matrix | Figs 1–4 | bit.ly/spmVid | bit.ly/spmApp |
| Litre plot | Figs 5–8 | bit.ly/litreVid | bit.ly/litreApp |
| Volcano plot | Figs 9–12 | bit.ly/volcVid | bit.ly/volcApp |
| Parallel coordinate | Figs 13–16 | bit.ly/pcpVid | bit.ly/pcpApp |

**Table 2. Examples of independent layers of interactivity.**

| Plot | Layer | Geom-drawing interactivity | Geom-manipulation interactivity |
|------|-------|----------------------------|----------------------------------|
| Scatterplot matrix | Background | None | User hovers over background hexagons to view gene counts |
| | Foreground | User clicks on background hexagon to draw corresponding genes as foreground points. Background layer does not need to be redrawn | User hovers over foreground points to view gene names |
| Litre plot | Background | User uses Shiny buttons to specify treatment pairs and hexagon sizes for drawing background hexagons | User hovers over background hexagons to view gene counts |
| | Foreground | User uses Shiny buttons to specify metric, metric order, and point size for drawing foreground points. Background layer does not need to be redrawn | User hovers over foreground points to view gene names |
| Volcano plot | Background | User uses Shiny buttons to specify treatment pairs and hexagon sizes for drawing background hexagons | User hovers over background hexagons to view gene counts |
| | Foreground | User uses Shiny buttons to specify point size, log fold changes, p-values to draw foreground points. Background layer does not need to be redrawn | User hovers over foreground points to view gene names |

technology can enhance the exploration of large datasets, especially in cases where one layer contains large amounts of data (such as the full dataset) and the other layer contains smaller amounts of data (such as a data subset). Because the layers are independent, users can save time and computation by keeping the layer with more data unaltered while only redrawing the layer with less data.

The idea of multi-layered interactive graphics in R was raised in [16] and a pilot implementation was made using qt software to create different listeners for different layers, an approach that is no longer functional. While the concept of independent layers of interactivity in R is not new in itself, the technology we introduce is new and solves a difficult problem that has been raised before. We achieved our independent double-layered interactivity using htmlwidgets [17], ggplot2 [18], shiny [19], JavaScript, and plotly [20].

In general, we first converted a static ggplot2 object into an interactive plotly object using ggplotly(). To overlay an additional layer of interactivity to the plotly object, we used the onRender() method of the htmlwidgets package [17]. This method allowed us to overlay an interactive foreground layer via plotly traces while the original interactive background layer of the plotly object did not need to be redrawn, something that cannot foreseeably be achieved with the native onRender() method of the plotly package [20]. Specifically, the htmlwidgets

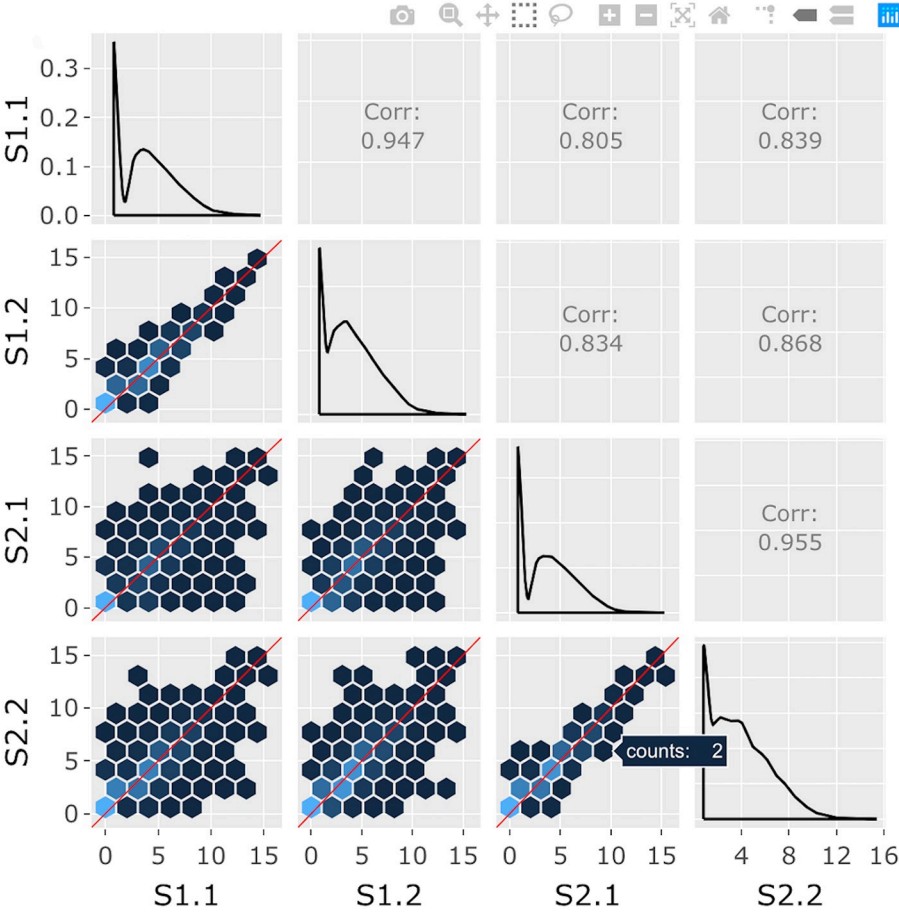

**Fig 1. Step 1: Independent interactive layers of scatterplot matrix.** First step in a four-part example series of user actions. User hovers over background hexagon to determine it contains two genes.

onRender() method contains three input parameters: an HTML Widget object, a character vector containing JavaScript code, and a list of R objects that can be serialized to JSON format. To develop our technique, we specified our plotly object as the HTML Widget object, which allowed for an interactive background. Within the method, we wrote JavaScript code that enabled interactive foregrounds to be updated without redrawing the interactive background. We used the R object list to transfer count tables and DEG lists into the method. In some our our applications, users can link between the layers of different interactive plots. This functionality was achieved by sending custom messages between the Shiny software and the JavaScript code within the htmlwidgets method [19].

The current paper can benefit biology researchers and developers alike. Developers who would like to modify our code can access pseudocode and documented code separately in the supplementary material (see S1 Table). For biology researchers who would like to readily apply our visualization tools to their data, didactic materials (figures, applications, and videos) are available for each plotting type in Table 1. We will now briefly explain how our two-layered

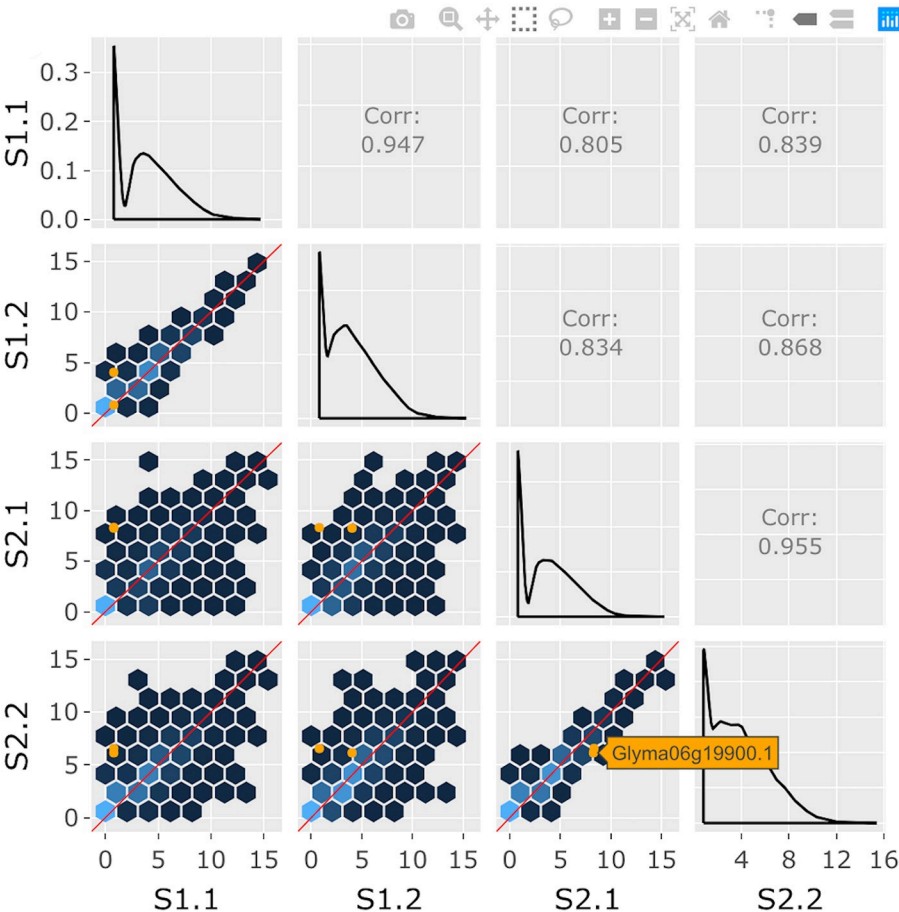

**Fig 2. Step 2: Independent interactive layers of scatterplot matrix.** Second step in a four-part example series of user actions. User clicks on background hexagon to overlay the two corresponding genes as orange points in the foreground layer of each scatterplot. The computationally-expensive background layer of hexagons does not need to be redrawn.

interactivity method can improve upon several of the RNA-seq visualization tools in our package.

**Scatterplot matrices.** Scatterplot matrices have appeared in statistical graphics literature for almost four decades and used across various fields of multivariate research [21–24]. Previous user studies have shown that participants performed better when using animated rather than static versions of scatterplot matrices. Users also preferred animated scatterplot matrices and found them easier to understand as they can alleviate overplotting issues [25]. Rendering scatterplot matrices interactive is promising but challenging with large datasets [26]. The number of background geoms that need to be drawn grows exponentially by dimension size: $n$-dimensional data corresponds to $n^2$ scatterplots. Our two-layered interactive visualization technology improves upon this dilemma by allowing details of interest to be superimposed in the foreground while the massive number of geoms in the background does not require redrawing. See Tables 1 and 2 and Figs 1–4 for details about our interactive scatterplot matrices.

**Litre plots.** Problems still remain when scatterplot matrices are applied to large datasets. Physical space requirements increase exponentially. Hence, when extended to large

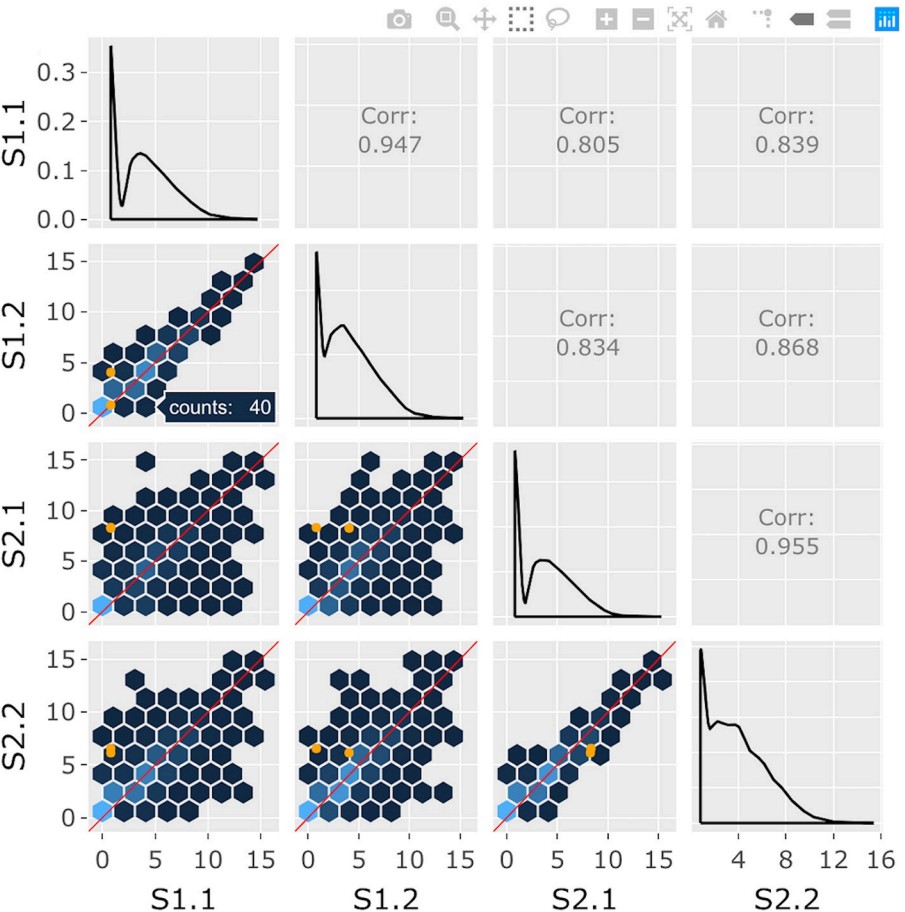

**Fig 3. Step 3: Independent interactive layers of scatterplot matrix.** Third step in a four-part example series of user actions. The background layer of hexagons remains interactive and the user can still hover over another hexagon of interest to determine it contains 40 genes.

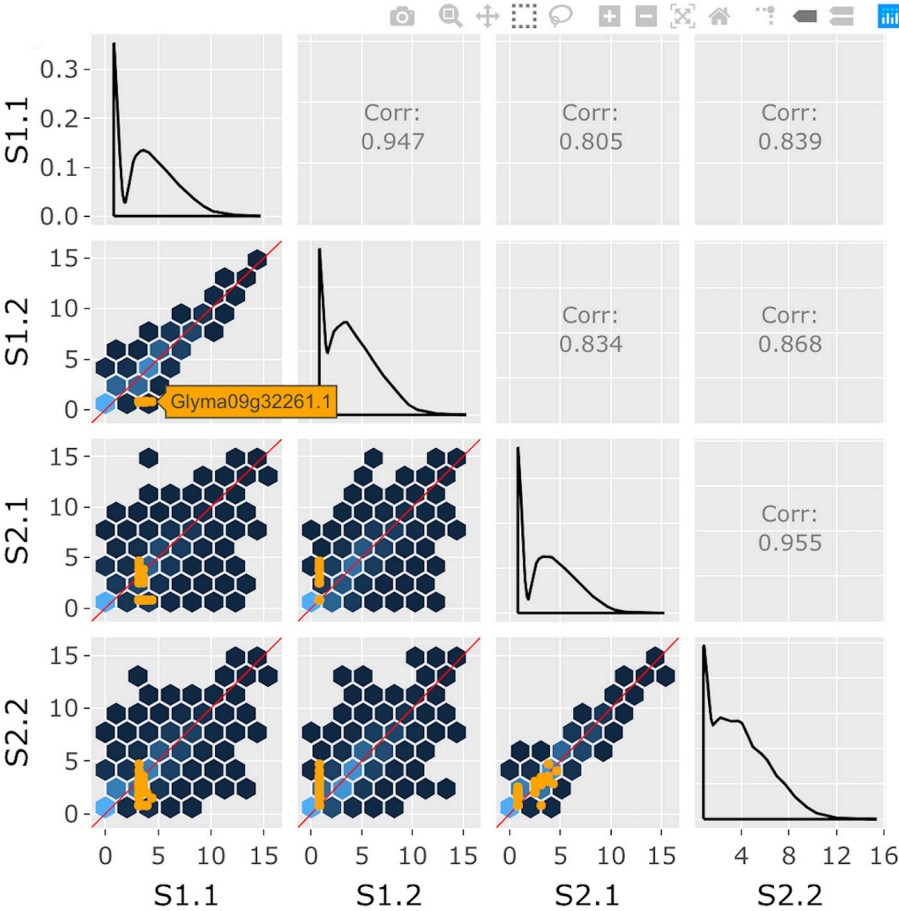

**Fig 4. Step 4: Independent interactive layers of scatterplot matrix.** Fourth step in a four-part example series of user actions. User clicks on background hexagon to overlay the 40 corresponding genes as orange points in the foreground layer of each scatterplot. This step does not require the computationally-expensive background layer of hexagons to be redrawn.

dimensions, it becomes difficult to mentally link many small plots within the matrix [27]. Several techniques have been proposed to ameliorate this problem. Three dimensional scatterplots are useful but can cause occlusion and depth perception issues [27]. Other techniques like grand tours [28], projection pursuits [29, 30], and scagnostics [31] have been proposed.

Even though these alternative techniques are useful, they may not simultaneously display distributions across all cases (genes) and variables (samples). We generally want to compare replicate and treatment variability in RNA-seq data, which can be visually accomplished by plotting all genes and samples. We also want to superimpose DEGs to determine how their read count variability compares to that of the whole dataset. In light of this, we developed a plot that collapses the scatterplot matrix onto one Cartesian coordinate system, allowing users to visualize all read counts from one DEG of interest onto all read counts of all genes in the dataset. We call this new plot a repLIcate TREatment ("litre") plot. An in depth explanation about the litre plot can be found in our previous methods paper [32].

We believe our two-layered interactive visualization method is an indispensable component of the litre plot. Drawing the background (all genes in the dataset) is the time-limiting step,

whereas drawing the foreground (one DEG of interest) is immediate. Most users would like to superimpose DEGs from a list one by one onto the background. This process would be unnecessarily time-prohibiting if the background needed to be redrawn each time the user progressed to the next DEG. Fortunately, our technology allows the user to immediately redraw the interactive foreground (the DEG of interest) while the background (all genes in the data) remains unchanged but preserved in its interactive capabilities. See Tables 1 and 2 and Figs 5–8 for details about our interactive litre plots.

**Volcano plots.** Volcano plots draw significance and fold change on the vertical and horizontal axes respectively. In RNA-seq studies, volcano plots allow users to check that genes were not falsely deemed significant due to outliers, low expression levels, and batch effects [33]. Researchers benefit from the ability to quickly identify individual gene names in the volcano plot. This was previously achieved with the identify() method in R, which identifies the closest point in a scatterplot to the position nearest the mouse click [33]. The interactive volcano plot in bigPint can identify individual gene names in a less ambiguous fashion by responding to users hovering *directly* over corresponding points. It also improves upon traditional volcano plots by allowing users to threshold on statistical values in order to immediately update the superimposed gene subset without having to redraw the more computationally-heavy background that contains all genes. See Tables 1 and 2 and Figs 9–12 for details about our interactive volcano plots.

**Consecutive box selection.** The bigPint package provides interactive tools for consecutive box selection. A box selection is a rectangular query drawn directly on a two-dimensional

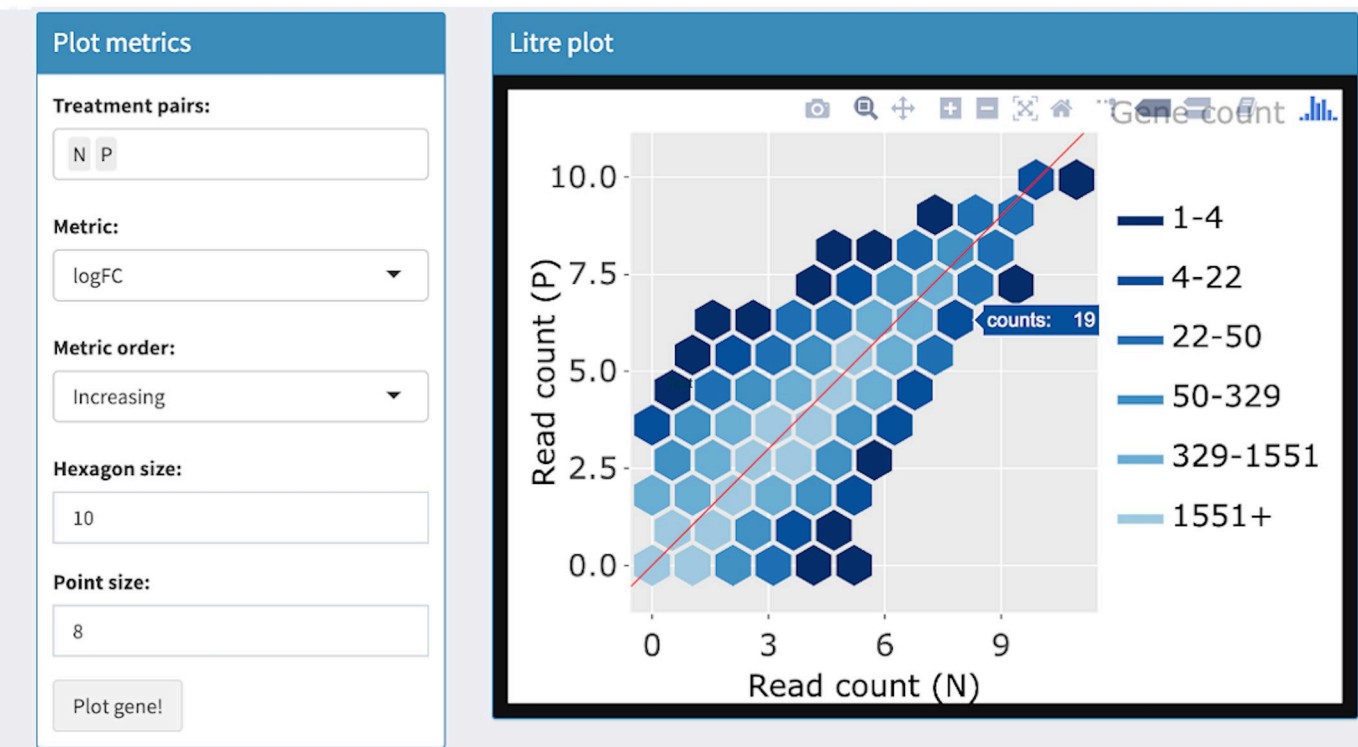

**Fig 5. Step 1: Independent interactive layers of litre plot.** First step in a four-part example series of user actions. User uses Shiny buttons to specify treatment pairs (N and P) and hexagon size (10) for drawing background hexagon layer. User can hover over hexagon of interest to determine it contains 19 genes.

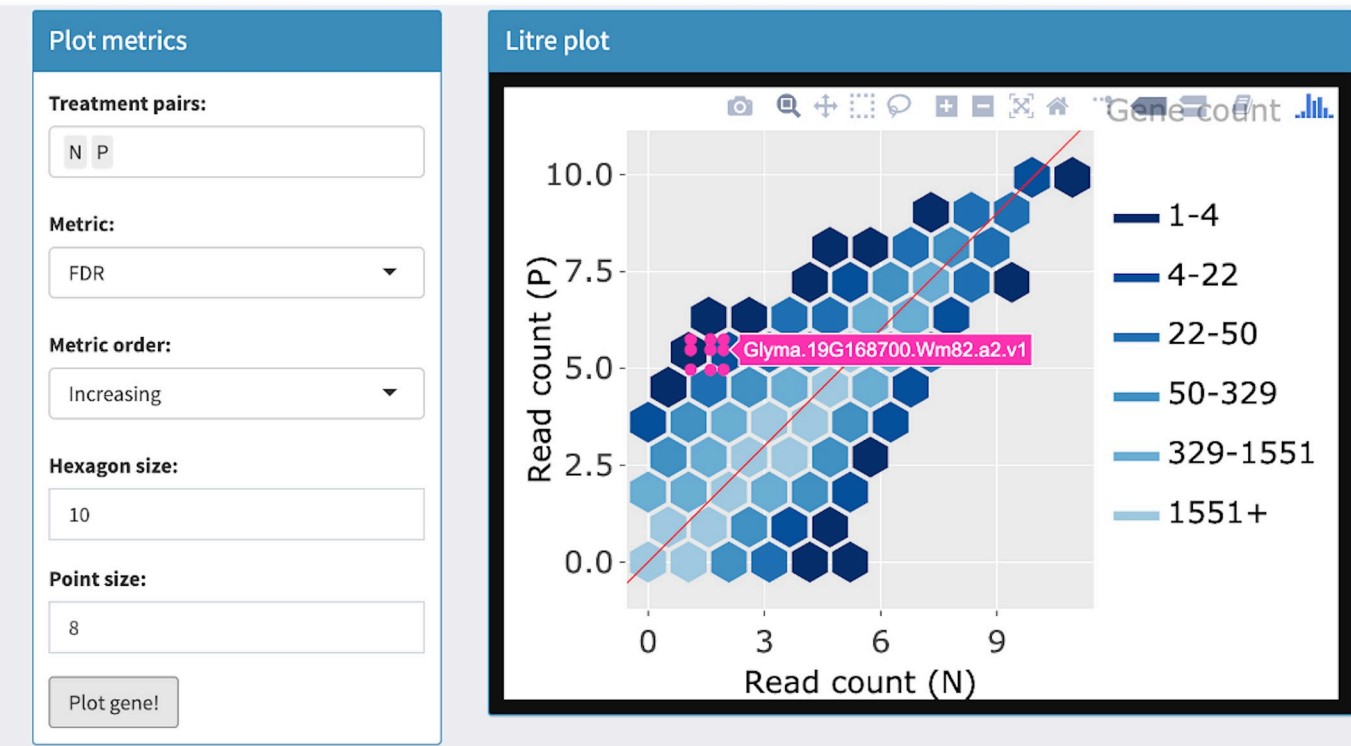

**Fig 6. Step 2: Independent interactive layers of litre plot.** Second step in a four-part example series of user actions. User uses Shiny buttons to specify metric (FDR) and metric order (Increasing) to establish the order in which genes will be overlaid as pink points in the foreground layer. User clicks "Plot gene" button and the gene with the lowest FDR value (Glyma.19G168700.Wm82.a2.v1) is overlaid. The background layer of hexagons does not need to be redrawn.

graph. Users can specify a box selection by clicking on the desired starting point of the rectangular query and dragging the mouse pointer to the desired opposite corner point of the rectangular query. This procedure for generating rectangles is widely used in interactive programs and should be familiar to most users [34]. After the user releases the mouse, the query is processed and only the data cases that were inside the specified rectangle remain. More precisely, a data case remains in a box selection queried between $(x_1, y_1)$ and $(x_2, y_2)$ if every point within $x_1 \leq x \leq x_2$ is also within $y_1 \leq y \leq y_2$ (where $y_2 \geq y_1$ and $x_2 \geq x_1$). The user can specify consecutive queries with multiple box selections. The consecutive box selection model is convenient in cases where identical thresholds are desired over adjacent features. In these cases, a single box selection of width $w$ can be used to simultaneously query the same threshold across $w$ features. This process is an improvement over single-feature box selection widgets, where $w$ individual queries would be required [34].

Consecutive box selection may have originally been designed for time series data, but has since proven useful for detecting patterns in gene expression data. Combined with parallel coordinate plots, the consecutive box selection technique has been used to elicit candidate regulatory splice sequences showing high values at some positions and low values at other positions [34]. In RNA-seq, this technology can also be used to investigate differential expression showing high read counts for one treatment group and low read counts for another treatment group, requiring a consecutive query. Consecutive box selection tools have been published for gene expression analysis sofware that was restricted for certain operating systems [34]. We

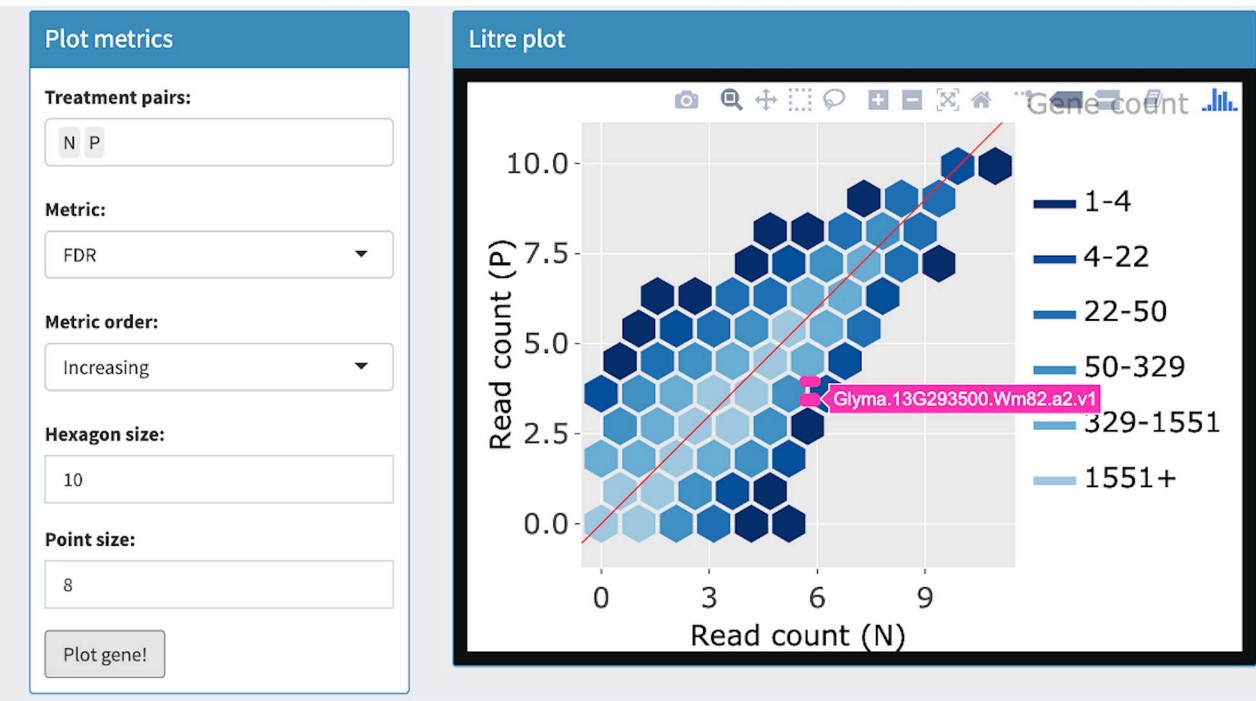

**Fig 7. Step 3: Independent interactive layers of litre plot.** Third step in a four-part example series of user actions. User clicks "Plot gene" button again and the gene with the second-lowest FDR value (Glyma.13G293500.Wm82.a2.v1) is overlaid. This step does not require the background layer of hexagons to be redrawn.

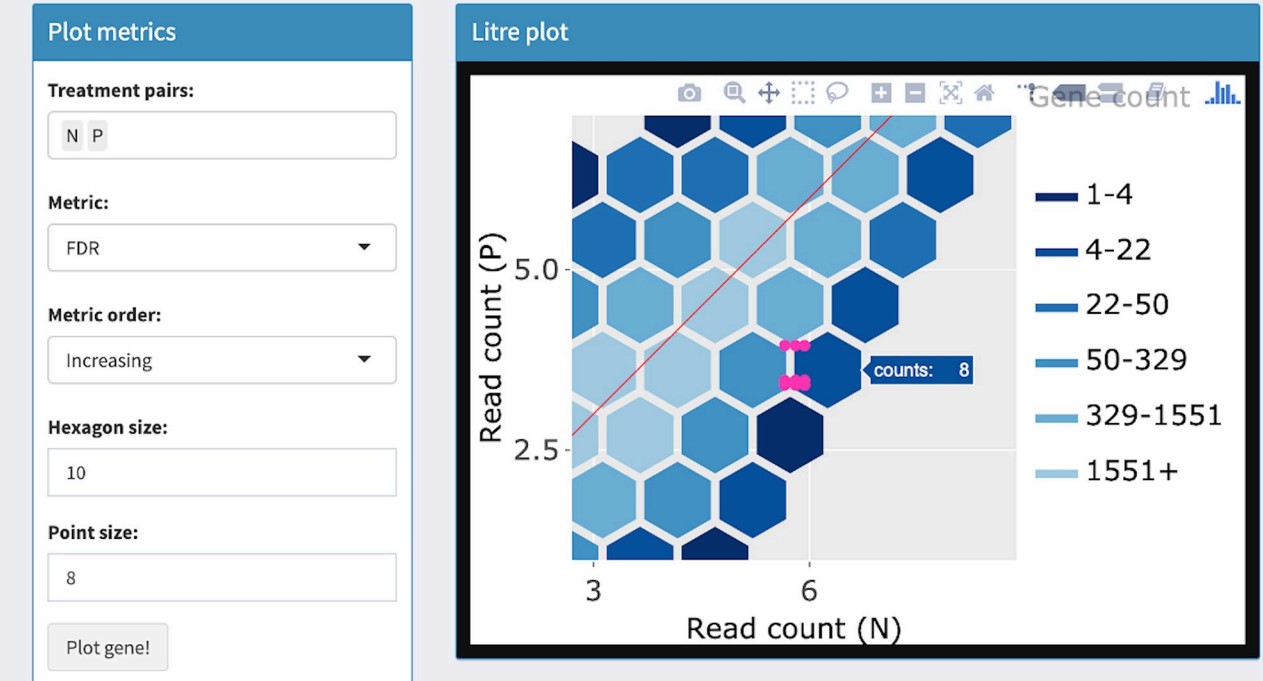

**Fig 8. Step 4: Independent interactive layers of litre plot.** Fourth step in a four-part example series of user actions. User can zoom and pan on the layers using the Plotly Modebar.

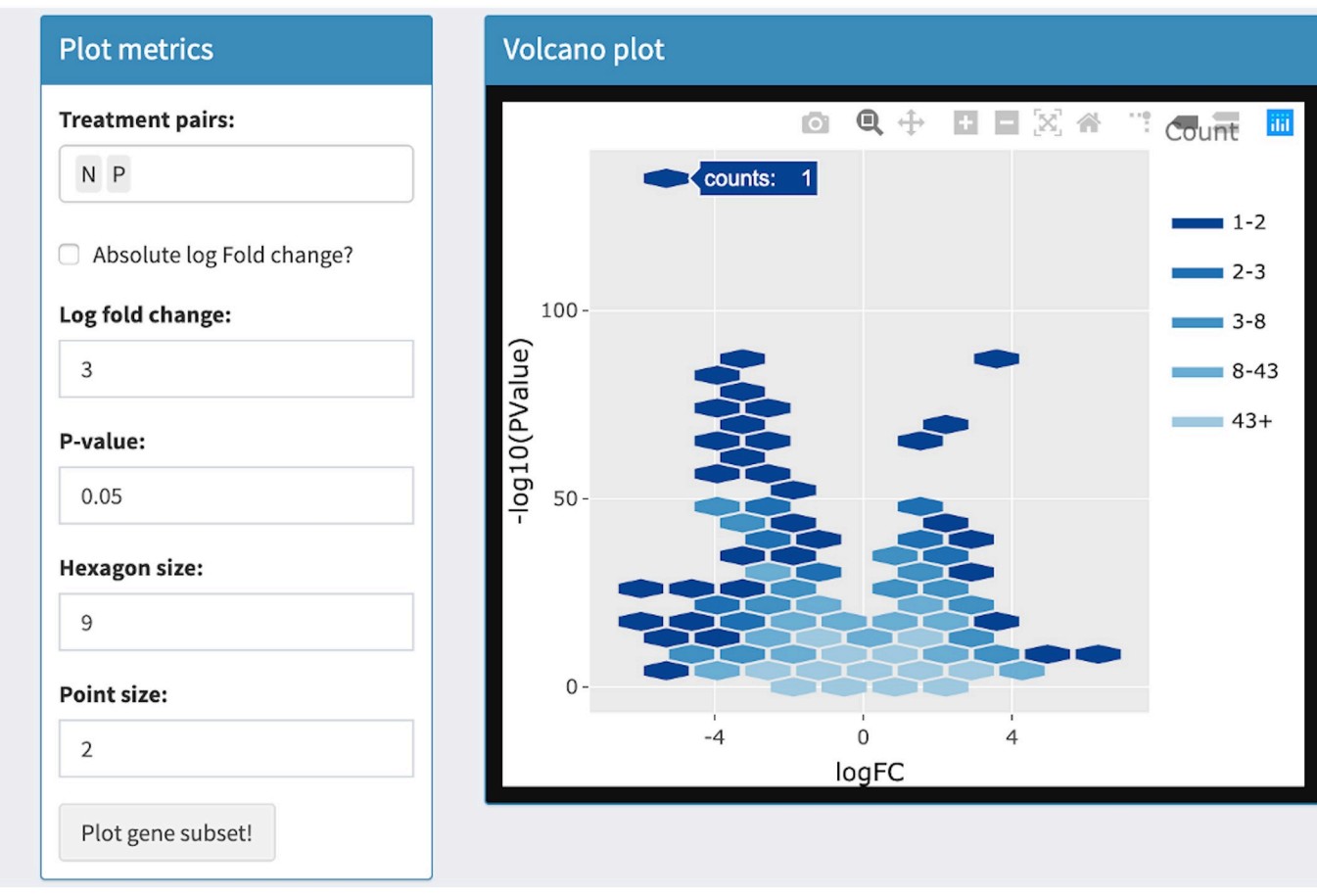

**Fig 9. Step 1: Independent interactive layers of volcano plot.** First step in a four-part example series of user actions. User uses Shiny buttons to specify treatment pairs (N and P) and hexagon size (9) for drawing background hexagon layer. User can hover over hexagon of interest to determine it contains 1 gene.

believe that publishing consecutive box selection tools in a platform like R can be useful for computational biologists using various operating systems. See Tables 1 and 2 and Figs 13–16 for details about our interactive parallel coordinate plots that feature consecutive box selection.

## Useful features

**Tailoring and saving static plots.** Static plots can be saved as list objects in the R workspace and/or as JPG files to a directory chosen by the user. Saving plots into the R workspace allows users to integrate them into analysis workflows. It also allows them to tailor the plots (such as adding titles and changing label sizes) using the grammar of graphics via the conventional + syntax. Saving plots to a directory allows users to keep professional-looking files that can be inserted into proposals and talks. By default, the bigPint package saves static plots both in the R workspace and a directory (the default location is tempdir()).

**Second feature layer.** Both static and interactive plots allow for a subset of data to be plotted in a different manner than the full dataset. When analyzing RNA-seq data, this second feature layer could represent DEGs. There are three options for creating data subsets with static

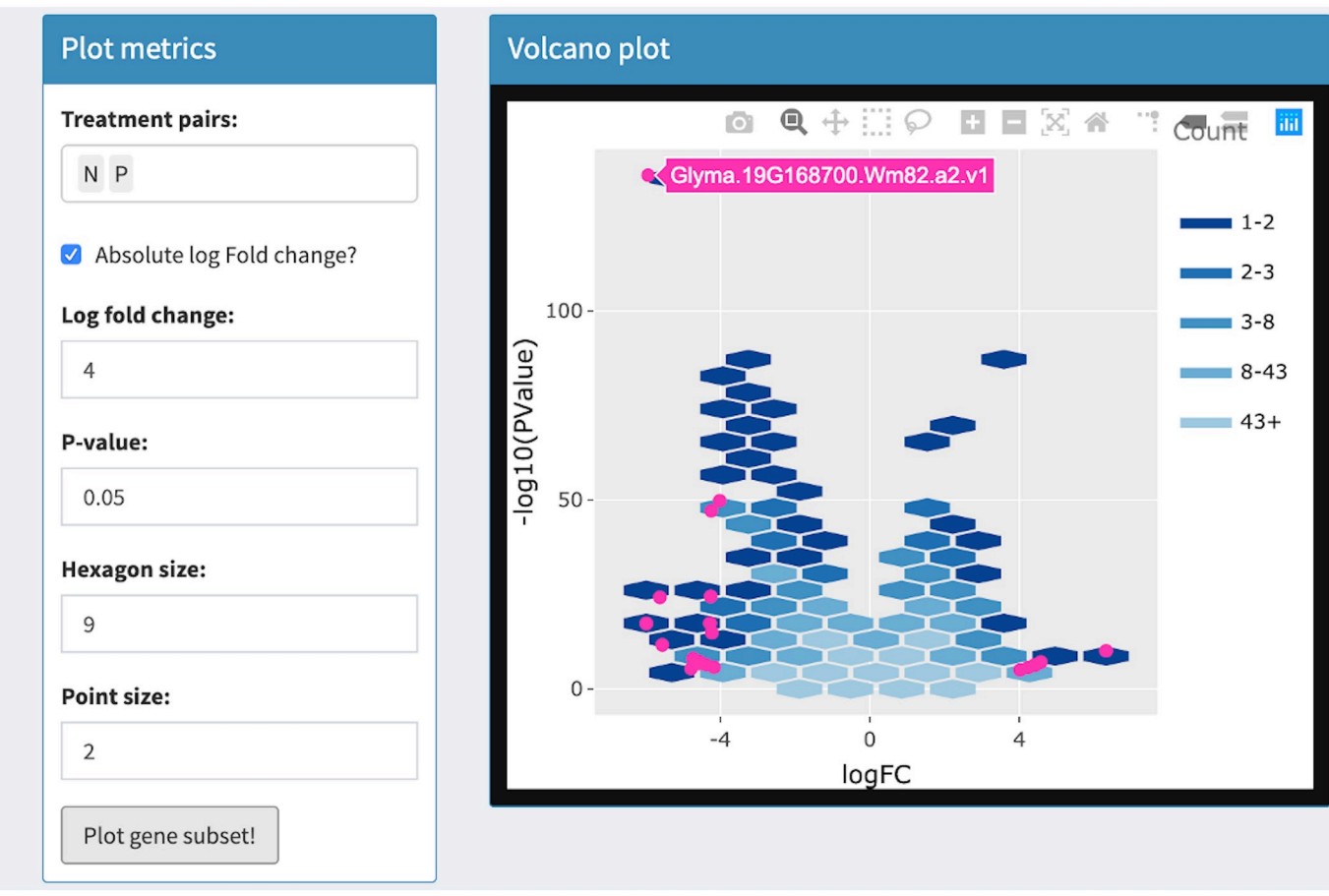

**Fig 10. Step 2: Independent interactive layers of volcano plot.** Second step in a four-part example series of user actions. User uses Shiny buttons to specify log fold change and p-value thresholds. User clicks "Plot gene subset" button and the subset of genes that pass the thresholds are overlaid in the foreground layer as pink points. The background layer of hexagons does not need to be redrawn. User hovers over foreground point to view gene name (Glyma.19G168700.Wm82.a2. v1).

plots. First, users can threshold the previously-mentioned *dataMetrics* object by one of its quantitative variables. Second, users can simply declare a geneList object that contains the list of data subset IDs. Third, the user can simply leave the *dataMetrics* and geneList objects to their default value of NULL and not overlay any data subsets.

**Group comparison filters.** When users create static plots, the package automatically creates a separate plot for each pairwise combination of treatment groups from the inputted data. When users explore interactive plots, fields are dynamically generated from the inputted data so that any pairwise combination of treatment groups can be selected by buttons. Users can then quickly flip between contrasts in their data. The bigPint package comes with an example soybean cotyledon dataset that has three treatment groups, which is used across several easy-to-follow articles on the package website. These assets can assist users who have data containing more than two treatment groups.

**Hexagonal binning.** Most bigPint plots represent genes using point geoms (where each point represents one gene) or hexagonal binning geoms (where each hexagon color represents the number of genes in that area). Plotting each gene as a point allows for ideal levels of detail

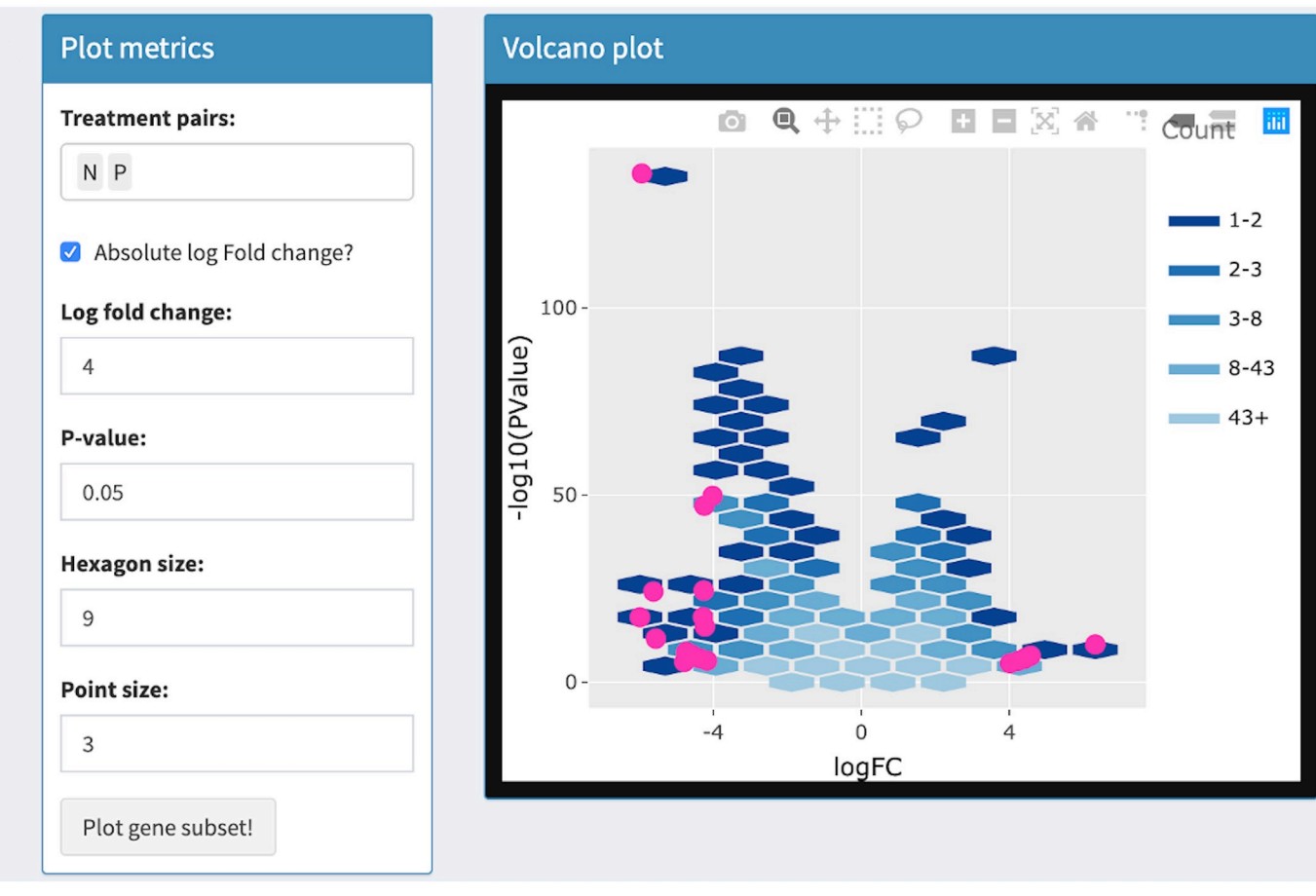

**Fig 11. Step 3: Independent interactive layers of volcano plot.** Third step in a four-part example series of user actions. User uses Shiny buttons to increase point size from 2 to 3. Foreground layer of pink points are increased in size and the background layer of hexagons does not need to be redrawn.

but overplotting can occur as the data increases, which makes it difficult to determine how many genes are in a given area. Hexagonal binning has been used in prior software to successfully manage overplotting issues [26, 35] and has shown superior time performance because less geom objects need to be plotted. The bigPint package allows users to draw the background using either geom, as preferences can depend on the dataset.

**Hierarchical clustering.** Users can conduct hierarchical clustering analyses on their data using the function plotClusters(). By default, the resulting clusters will be plotted as parallel coordinate lines superimposed onto side-by-side boxplots that represent the five-number summary of the full dataset. There are three main approaches in the plotClusters() function:

- Approach 1: The clusters are formed by clustering only on a user-defined subset of data (such as significant genes). Only these user-defined genes are overlaid as parallel coordinate lines.

- Approach 2: The clusters are formed by clustering the full dataset. Then, only a user-defined subset of data (such as significant genes) are overlaid as parallel coordinate lines.

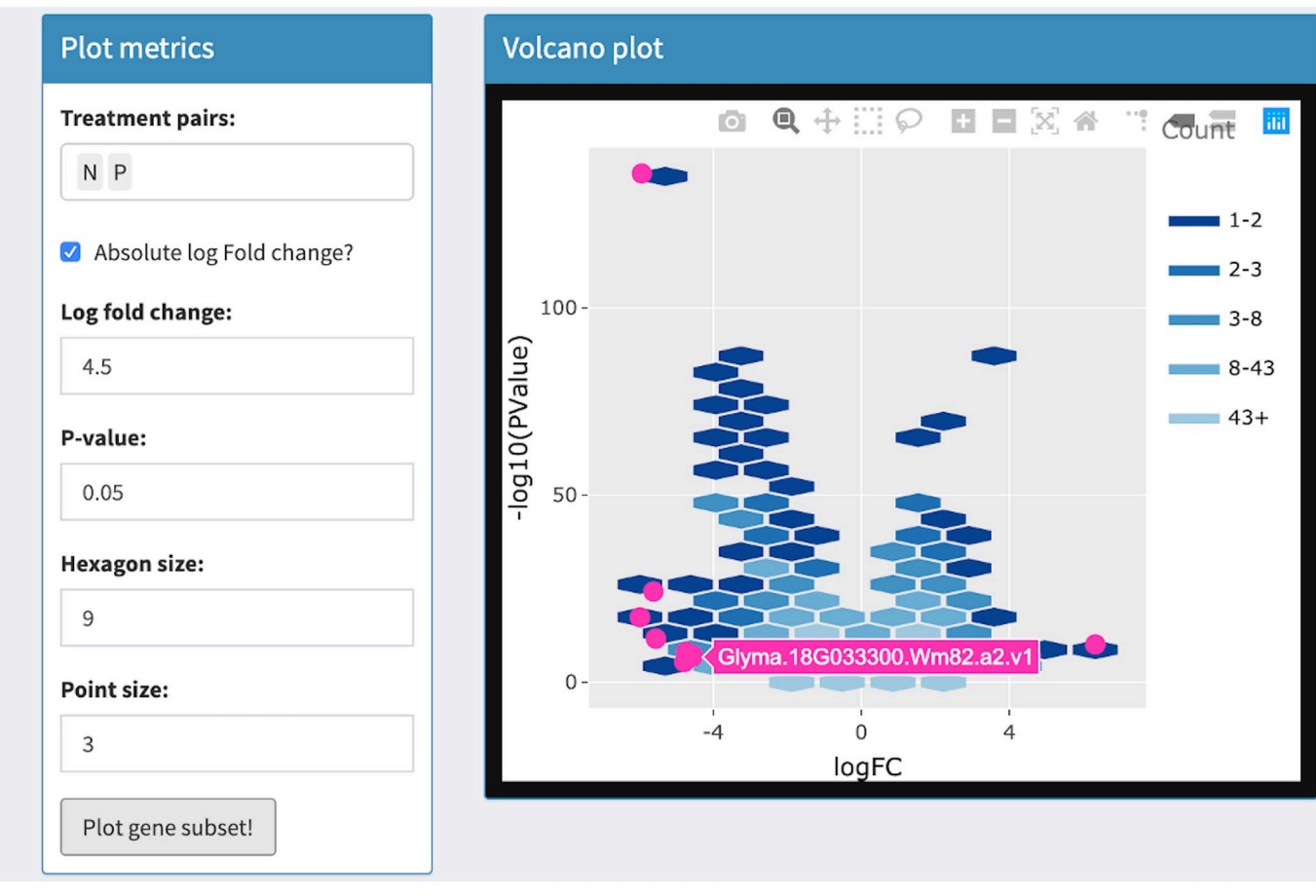

**Fig 12. Step 4: Independent interactive layers of volcano plot.** Fourth step in a four-part example series of user actions. User uses Shiny buttons to update threshold values and again presses "Plot gene subset" button. The subset of genes that pass the new thresholds are overlaid in the foreground layer as pink points. The background layer of hexagons does not need to be redrawn.

- Approach 3: The clusters are formed by clustering the full dataset. All genes are overlaid as parallel coordinate lines.

The clustering algorithm is based on the hclust() and cutree() functions in the R stats package. It offers the same set of agglomeration methods ("ward.D", "ward.D2", "single", "complete", "average", "mcquitty", "median", and "centroid") with "ward.D" as the default. In many cases, users may want to save clusters derived from the plotClusters() function for later use, such as to overlay them onto scatterplot matrices, litre plots, and volcano plots. The gene IDs of each cluster can be saved as .RDS files for this purpose by setting the verbose option of the plotClusters() function to a value of TRUE.

**Various plot aesthetics.** Users can modify various aesthetics for both static and interactive plots, including geom size. Some plots also provide alpha blending, which can benefit users plotting large datasets as parallel coordinate lines [36]. Statistical coloring is inconsistent in numerous packages even though it can greatly enhance biological data visualization [37]. The bigPint package allows users to maintain consistent coloring across hierarchical clusters and when working between various plots.

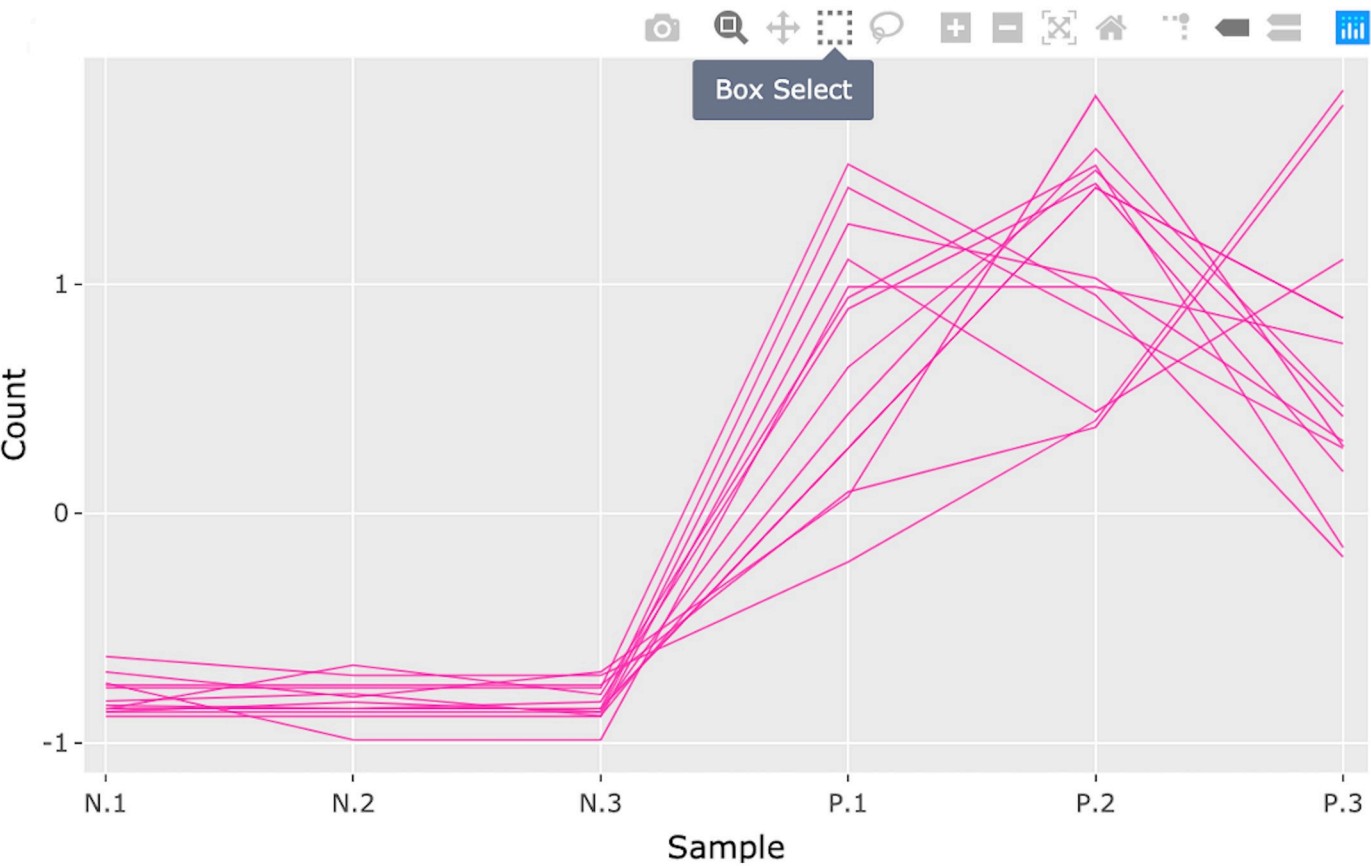

**Fig 13. Step 1: Consecutive box selection in parallel coordinate plot.** First step in a four-part example series of user actions. User selects the Box Select tool from the Plotly Modebar.

**Selection and aggregation.** Some techniques that are effective in data exploration may lose their efficiency and eventually fail as data size increases. Two main approaches to solving this problem are data selection and data aggregation [38]. Data selection means that only a subset of the full data is displayed at a given time. The data subset can be selected through queries and interactive controls which allow the user to quickly examine different data subsets [38]. Data aggregation means that the full dataset is divided into data subsets (called aggregates) that reduce the amount of data being simultaneously visualized. Users with large datasets should ideally be able to perform both data selection and data aggregation [38]. The bigPint package allows users to easily perform data selection using queries (such as thresholds and sliders) and interactive controls (such as zooming, box and lasso selection, and panning) and to perform data aggregation using hierarchical clustering.

**Shiny interactivity.** Interactive plots in the bigPint package open as Shiny applications that consist of simple dashboards with "About" tabs that explain how to use the applications. They also include "Application" tabs that provide several input fields for the user to tailor their plots. Some of these input fields are generated dynamically from the inputted dataset so that users have more convenience in how they select data subsets. In these applications, users can also download lists of selected genes and static images of interactive graphics to their local computers.

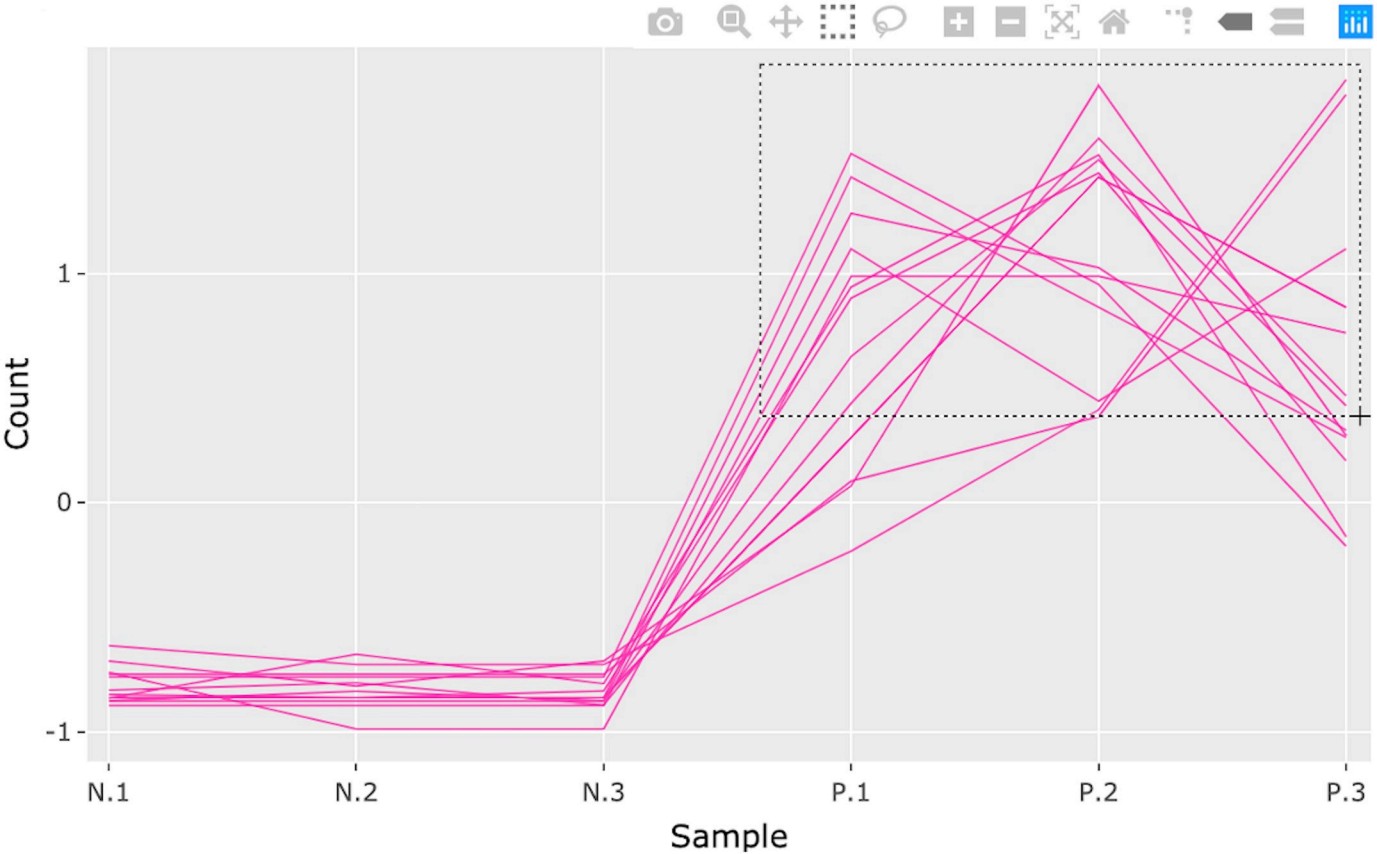

**Fig 14. Step 2: Consecutive box selection in parallel coordinate plot.** Second step in a four-part example series of user actions. User specifies box selection by drawing a rectangular query. Only the genes (pink lines) inside the specified rectangle remain.

Shiny applications can be launched on a local personal computer, hosted on a local or cloud-based server, or hosted for free on the shinyapps.io website. As such, interactive big-Pint packages can be deployed on a personal computer using only a local file containing the data, the bigPint package and its dependencies, R / RStudio, and a browser recommended by Shiny (Google Chrome or Mozilla Firefox). This method does not require internet connectivity, which can be useful for users who are protecting sensitive data, analyzing or presenting data in contexts without reliable connectivity, or testing and developing applications.

## Results

The bigPint package contains scatterplot matrices, volcano plots, litre plots, and parallel coordinate plots as example graphics that implement our new layered interactivity technology. In a recent methods paper, we used public RNA-seq datasets to demonstrate how these particular bigPint graphics can help biologists detect crucial issues with normalization methods and DEG designation in ways not possible with numerical models [32]. We also applied these big-Pint visualization tools in a recent research paper that sought to elicit how nutrition and viral infection affect the honey bee transcriptome [39].

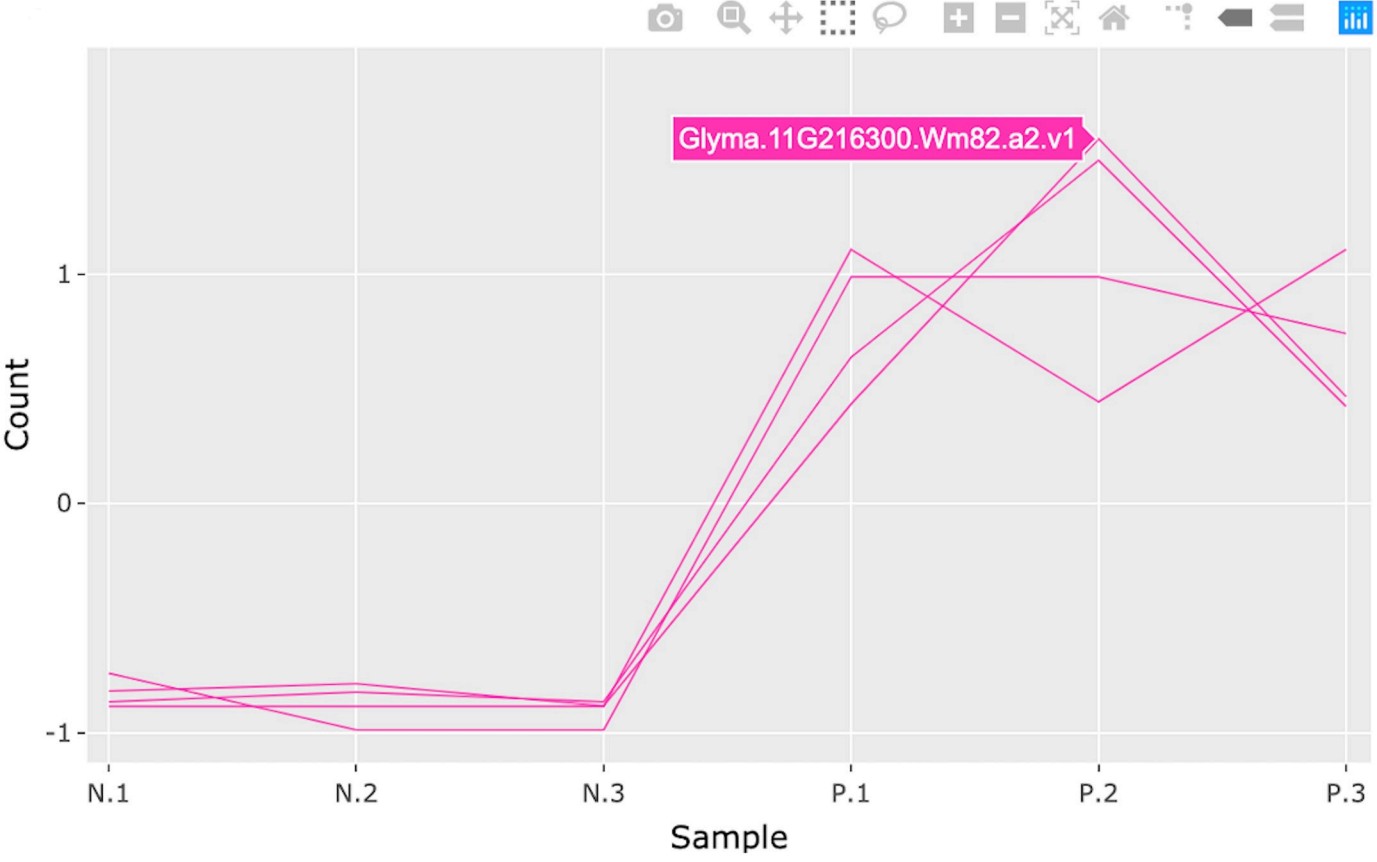

**Fig 15. Step 3: Consecutive box selection in parallel coordinate plot.** Third step in a four-part example series of user actions. User can hover over a gene of interest (pink line) to view its name (Glyma.11G216300.Wm82.a2.v1).

## Availability and future directions

The four public datasets used in our recent methods paper [32] are available online: Three are deposited on the NCBI Sequence Read Archive with accession numbers SRA000299 [40], PRJNA318409 [41], and SRA048710 [42]. One is deposited on the NCBI Gene Expression Omnibus with accession number GSE61857 [43]. The original dataset we used in our recent methods paper [39] is also available on the NCBI Gene Expression Omnibus with accession number GSE121885.

The bigPint package itself comes with two of the aforementioned RNA-sequencing datasets as examples [42, 43]. The package can be downloaded from the Bioconductor website (https://bioconductor.org/packages/devel/bioc/html/bigPint.html). As linked in the Bioconductor website, the bigPint package also has a vignette website (https://lindsayrutter.github.io/bigPint), where users can follow reproducible code to install the software and use the example datasets to create bigPint graphics and follow an example analysis pipeline. Users can also report bugs and submit requests.

In this paper, we introduced new visualization tools that enable independent layers of interactive capabilities for the foreground and background of plots using Plotly in R. We believe this methodology represents a fairly novel contribution to the field of interactive data visualization that can lead to sizable performance gains when working with large datasets. Advocating

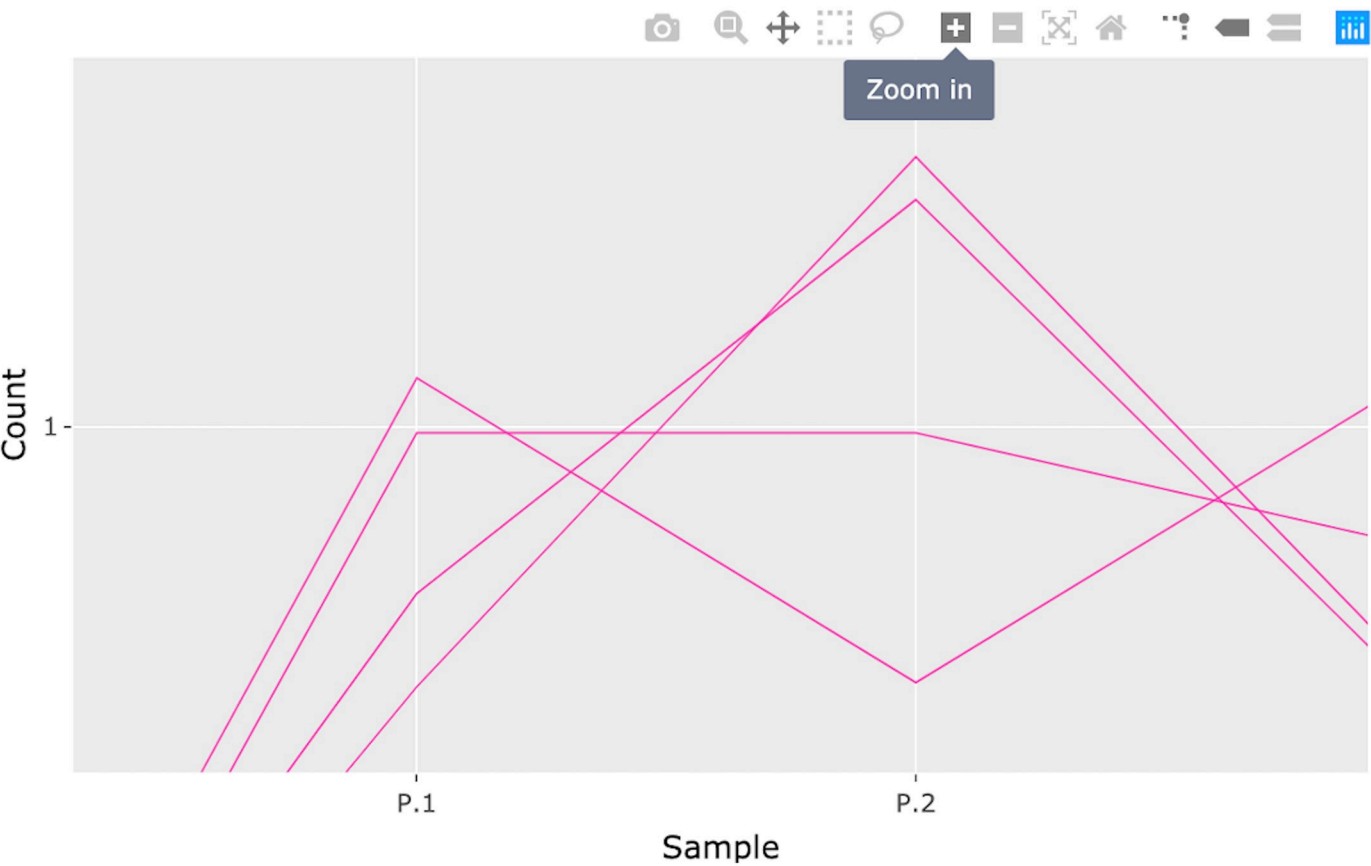

**Fig 16. Step 4: Consecutive box selection in parallel coordinate plot.** Fourth step in a four-part example series of user actions. User can zoom and pan on the plot using the Plotly Modebar.

state-of-the-art visualization tools is crucial for biology researchers to analyze and present their data and for visualization researchers to develop novel methods. We anticipate that our documented source code and pseudocode may encourage computational scientists to expand upon our layered interactive technology and/or apply it in new ways toward other computational biology tasks.

## Supporting information

**S1 Table. Resources for developers about bigPint interactive graphics.**
(PDF)

**S1 Pseudocode. Pseudocode for interactive scatterplot matrix.**
(PDF)

**S2 Pseudocode. Pseudocode for interactive litre plot.**
(PDF)

**S3 Pseudocode. Pseudocode for interactive volcano plot.**
(PDF)

**S4 Pseudocode. Pseudocode for interactive parallel coordinate plot.**
(PDF)

## Acknowledgments

We would like to thank Roxane Legaie (Senior Bioinformatician at Monash University at the time) and Dr. Amy L. Toth (Associate Professor in the Departments of Entomology and Ecology, Evolution, and Organismal Biology at Iowa State University) for their helpful feedback. We also thank the open-source community for their numerous outlets of support in regards to R, plotly, htmlwidgets, ggplot2, and shiny, without which this project would not have been possible.

## Author Contributions

**Conceptualization:** Lindsay Rutter, Dianne Cook.

**Software:** Lindsay Rutter.

**Supervision:** Dianne Cook.

**Visualization:** Lindsay Rutter.

**Writing – original draft:** Lindsay Rutter.

**Writing – review & editing:** Lindsay Rutter, Dianne Cook.

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
