## [Decision Letter · Decision Letter 0]

18 Nov 2019

Dear Dr Rutter,

Thank you very much for submitting your manuscript 'bigPint: A Bioconductor visualization package that makes big data pint-sized' for review by PLOS Computational Biology. Your manuscript has been fully evaluated by the PLOS Computational Biology editorial team and in this case also by independent peer reviewers. The reviewers appreciated the attention to an important problem, but raised some substantial concerns about the manuscript as it currently stands. While your manuscript cannot be accepted in its present form, we are willing to consider a revised version in which the issues raised by the reviewers have been adequately addressed. We cannot, of course, promise publication at that time.

Sincerely,

Aaron E. Darling

Software Editor

PLOS Computational Biology

[LINK]

Editor's specific comments:

thank you for your patience while this manuscript was under review, as there was difficulty sourcing appropriately qualified reviewers. Please thoroughly address the comments of both reviewers in your revisions, in particular those focused on software architecture, reliability, and ease of use.

Reviewer's Responses to Questions

**Comments to the Authors:**

Reviewer #1: Rutter and Cook discuss their new R package offering data visualization techniques geared towards RNA-seq data. The data visualization techniques make clever use of interactivity to highlight aspects of the data, which enable researchers to identify common analysis errors. While I like the concept of the presented visualization techniques, I am not convinced that this will be a widely applied tool. The tool felt clunky and still produced several errors. I will outline these as well as other issues/suggestions below:

1. The authors claim that their software can be used for the exploration of any large biological datasets. However, their choice of plots, in particular the volcanoplot, suggests to me a focus on RNA-seq data. I would suggest the inclusion of another example for a non RNA-seq dataset or making it clear that this package targets RNA-seq data exclusively.

2. It is great that all plots are easily downloadable, but for the purpose of replicability it would be great if logs of all user interactions could be provided. This can, for example, be handled with shinylogs (https://github.com/dreamRs/shinylogs).

3. Part of why I think the package feels clunky is that each plot is provided in a separate application. I think it would be more user friendly to provide one app that has user-specified outputs. A brilliant example of this is the iSEE package.

4. Finally, I am unsure why the author chose the dataMetrics object as input, instead of the much more common SummarizedExperiment object. This has the advantage that it offers support for DelayedMatrices which are extremely useful when datasets are very large.

5. How does bigPint scale? Have you tried analyzing single cell RNA-seq datasets?

Errors:

1. The example interactive application for the volcano plot is broken.

2. None of the parallel coordinate plots show the correct labels for genes when hovering over the lines associated with genes.

Reviewer #2: This is supplied as a R-markdown to show what I have done to try to use bigPint.

I have also supplied the file as an attachment.

---

title: "review_bigPint"

author: "Paul Brennan"

date: "12/10/2019"

output: html_document

---

## General comments and recommendation

This is an interesting and worthwhile manuscript that shares some very nice and novel technical and visualisation strategies.

### Novelty in visualisation

I believe that the hexagon concept is being used in a novel way. I believe it is worthwhile for high density data. I believe that the approach should fit well into the Bioconductor methodology. It would be useful to see the package integrated into a workflow.

### Good points

I like the supporting website that the authors have provided.

I like the videos they have created. These make the concept understandable and a very positive way.

### Minor concerns

I have some minor concerns that I believe need to be addressed prior to publication.

My first an most important concern is that I have found it quite difficult to reproduce the workflows and examples using the links in the manuscipt.

I have focussed my efforts on trying to make the bigPint package work using the links provided in the manuscript. Sadly, I have found that quite challenging.

For some reason, and it may be a limitation of my technical skills, when I accessed the code shown in the links for Table 1, I could not make the code work.

The first link to https://github.com/lindsayrutter/bigPint/blob/master/inst/shiny-examples/plotSMApp/app.R points to a very complex example. I do not believe it constitutes a "Helpful resource". When I copy and paste this data into a new R-Studio Project, it is very difficult to get to work.

#### Trying to get Scatterplot Matrix App to work...

Here I am going to use the R Markdown file to catalogue my attempts.

Script from [here](bit.ly/spmCode)

```{r cut_and_paste_from_bit_ly_spmCode, include=FALSE, eval=FALSE}

# I have removed this from the review because it uses up too many words...

# trust me please, I have cut and pasted it verbatim.

```

When I run this, a window opens and then closes again. Interesting my data

is a Value - not Data in my Global Environment and it says "NULL (empty)"

Why? Well let's run the data acquisition by itself.

Annoyingly this is not a reproducible problem.

Sometimes it seems to work but I don't know why.

```{r test_data_download, include=TRUE}

data <- bigPint:::PKGENVIR$DATA

str(data)

```

Still NULL, sadly.

All the other code examples use the same code to get data in so I'm not going to try them here. I have tried them previously.

#### Where else can I go for help?

Let's check (Bioconductor)[https://www.bioconductor.org/packages/release/bioc/html/bigPint.html].

The page is a good starting point.

It has a green build which is a good sign.

Check out the (R-script)[https://www.bioconductor.org/packages/release/bioc/vignettes/bigPint/inst/doc/bioconductor.R]

Woops! It seems blank. A bit frustrating.

[HTML Vignette](https://www.bioconductor.org/packages/release/bioc/vignettes/bigPint/inst/doc/bioconductor.html)

points to website with a [rotten url](https://lindsayrutter.github.%20io/bigPint/)! Woops again!

Let's go to the Github Repository and see what we can find...

The [link](https://lindsayrutter.github.io/bigPint/) is in the manuscript

and works. Yeah!!

OK, now we have some different code:

```{r test_data_download, include=TRUE}

library(bigPint)

data("soybean_cn_sub")

soybean_cn_sub <- soybean_cn_sub[,1:7]

app <- plotSMApp(data=soybean_cn_sub)

if (interactive()) {

shiny::runApp(app)

}

```

Good new is that the data() function works and gives us some data!

```{r str_data}

str(soybean_cn_sub)

```

Nice data.frame produced.

Bad news is that my interactive plot still won't work!

Well actually it does work when I run this again.

It is a bit slow and it gives lots of red text:

'scatter' objects don't have these attributes: 't2'

What happens if I try the

```{r try_PKGENVIR$DATA_again}

data <- bigPint:::PKGENVIR$DATA

```

It works! If I do it after I run your App. Wonder why? I'm not going to spend any time working through that at the moment.

Now the script above works with lots of error messages.

#### Back to square one, can I make the static hex plots

```{r static_hex}

data(soybean_ir_sub)

soybean_ir_sub[,-1] <- log(soybean_ir_sub[,-1]+1)

data(soybean_ir_sub_metrics)

ret <- plotLitre(data = soybean_ir_sub,

dataMetrics = soybean_ir_sub_metrics, threshVal = 1e-10,

saveFile = FALSE)

length(ret)

names(ret)[1]

ret[[1]]

```

Answer is Yes! Good and it looks interesting with points highlighted on it.

I probably should have started here in the first place but hey...

#### I need to try to make the apps work...

I found some code here: [https://rdrr.io/bioc/bigPint/man/plotVolcanoApp.html]

```{r plot_Volcano_App}

library(bigPint)

# Example 1: Create interactive volcano plot of logged data using hexagon

# bins for the background.

data(soybean_cn_sub)

data(soybean_cn_sub_metrics)

app <- plotVolcanoApp(data = soybean_cn_sub,

dataMetrics = soybean_cn_sub_metrics)

if (interactive()) {

shiny::runApp(app)

}

```

OK, so this work. Excellent! I am reassured and in fact more than reassured.

I am pleased.

I can plot the gene subset and Download selected genes. I was under the impression that I would be able to click on a hexagon but I don't seem to be able to do that. Seems that functionality is not possible here.

```{r plot_Volcano_App}

# Example 2: Create interactive volcano plot of logged data using points for

# the background.

app <- plotVolcanoApp(data = soybean_cn_sub,

dataMetrics = soybean_cn_sub_metrics, option = "allPoints",

pointColor = "magenta")

if (interactive()) {

shiny::runApp(app)

}

```

This shows the points. The scale of the P-value seems a little unwise.

Should it really go all the way up to 1? I think there should be the option

to go down to lower P-values more easily and not up.

#### try PlotLitre App from online

```{r plot_Volcano_App}

data(soybean_ir_sub)

data(soybean_ir_sub_metrics)

soybean_ir_sub_log <- soybean_ir_sub

soybean_ir_sub_log[,-1] <- log(soybean_ir_sub[,-1]+1)

app <- plotLitreApp(data = soybean_ir_sub_log,

dataMetrics = soybean_ir_sub_metrics)

if (interactive()) {

shiny::runApp(app, port = 1234, launch.browser = TRUE)

}

```

I can see the number of genes in each hexagram if I hover over it - nice.

Plot gene works to give orange spots which can be hovered over to identify. However, I'm not sure how I am selecting those genes.

I have worked out that it is going through the genes by rank. However, because that information is off the bottom of the screen, it took me a while to realise.

I need to watch the video again!

Back to (website)[https://lindsayrutter.github.io/bigPint/articles/interactive.html]

Watch video about Scatterplot matrix app

Cut and paste code:

```{r plotSMAapp_again}

library(bigPint)

data("soybean_cn_sub")

soybean_cn_sub <- soybean_cn_sub[,1:7]

app <- plotSMApp(data=soybean_cn_sub)

if (interactive()) {

shiny::runApp(app)

}

```

This works! Excellent.

Selecting hexagrams works.

Downloading IDs works.

Downloading plots doesn't :-(

Need to open in Browser as advised!

Could add that as error message?

OK so it does work but it produces LOTS of warnings.

I wonder why?

Warning: 'scatter' objects don't have these attributes: 't2'

Lots of repeat of this warning.

#### Try plotLitreApp

```{r plotSMAapp_again}

data("soybean_ir_sub")

data("soybean_ir_sub_metrics")

soybean_ir_sub_log <- soybean_ir_sub

soybean_ir_sub_log[,-1] <- log(soybean_ir_sub[,-1]+1)

app <- plotLitreApp(data=soybean_ir_sub_log,

dataMetrics = soybean_ir_sub_metrics)

if (interactive()) {

shiny::runApp(app, port = 1234, launch.browser = TRUE)

}

```

#### Try plotPCPApp

```{r plotPCPApp}

soybean_ir_sub_st = as.data.frame(t(apply(as.matrix(soybean_ir_sub[,-1]), 1,

scale)))

soybean_ir_sub_st$ID = as.character(soybean_ir_sub$ID)

soybean_ir_sub_st = soybean_ir_sub_st[,c(length(soybean_ir_sub_st),

1:length(soybean_ir_sub_st)-1)]

colnames(soybean_ir_sub_st) = colnames(soybean_ir_sub)

nID = which(is.nan(soybean_ir_sub_st[,2]))

soybean_ir_sub_st[nID,2:length(soybean_ir_sub_st)] = 0

plotGenes = filter(soybean_ir_sub_metrics[["N_P"]], FDR < 0.01, logFC < -4) %>%

select(ID)

pcpDat = filter(soybean_ir_sub_st, ID %in% plotGenes[,1])

app <- plotPCPApp(data = pcpDat)

if (interactive()) {

shiny::runApp(app, display.mode = "normal")

}

```

Works!

In Browser, I can save images as advised.

Nice job.

#### Try Volcano app from website

Sadly no code on [website](https://lindsayrutter.github.io/bigPint/articles/interactive.html#volcano-plot-app) when accessed on 12 Oct 2019.

Example code from (here)[https://rdrr.io/bioc/bigPint/man/plotVolcanoApp.html] instead as above. Made it work again!

GREAT!

```{r session_info}

sessionInfo()

```

So, I made it work but not from code on the manuscript.

Please make it easier for me.

Here is my session info:

R version 3.6.1 (2019-07-05)

Platform: x86_64-apple-darwin15.6.0 (64-bit)

Running under: macOS Sierra 10.12.6

Matrix products: default

BLAS: /System/Library/Frameworks/Accelerate.framework/Versions/A/Frameworks/vecLib.framework/Versions/A/libBLAS.dylib

LAPACK: /Library/Frameworks/R.framework/Versions/3.6/Resources/lib/libRlapack.dylib

Random number generation:

RNG: Mersenne-Twister

Normal: Inversion

Sample: Rounding

locale:

[1] en_GB.UTF-8/en_GB.UTF-8/en_GB.UTF-8/C/en_GB.UTF-8/en_GB.UTF-8

attached base packages:

[1] stats graphics grDevices utils datasets methods

[7] base

other attached packages:

[1] shinycssloaders_0.2.0 Hmisc_4.2-0 Formula_1.2-3

[4] survival_2.44-1.1 lattice_0.20-38 RColorBrewer_1.1-2

[7] GGally_1.4.0 data.table_1.12.2 dplyr_0.8.3

[10] stringr_1.4.0 hexbin_1.27.3 tidyr_1.0.0

[13] htmlwidgets_1.5.1 plotly_4.9.0 ggplot2_3.2.1

[16] shinydashboard_0.7.1 shiny_1.4.0 bigPint_1.0.0

loaded via a namespace (and not attached):

[1] Rcpp_1.0.2 assertthat_0.2.1 zeallot_0.1.0

[4] digest_0.6.21 mime_0.7 R6_2.4.0

[7] plyr_1.8.4 backports_1.1.5 acepack_1.4.1

[10] httr_1.4.1 pillar_1.4.2 rlang_0.4.0

[13] lazyeval_0.2.2 rstudioapi_0.10 rpart_4.1-15

[16] Matrix_1.2-17 checkmate_1.9.4 labeling_0.3

[19] splines_3.6.1 foreign_0.8-72 munsell_0.5.0

[22] compiler_3.6.1 httpuv_1.5.2 xfun_0.10

[25] pkgconfig_2.0.3 base64enc_0.1-3 htmltools_0.4.0

[28] nnet_7.3-12 tidyselect_0.2.5 tibble_2.1.3

[31] gridExtra_2.3 htmlTable_1.13.2 reshape_0.8.8

[34] viridisLite_0.3.0 withr_2.1.2 crayon_1.3.4

[37] later_1.0.0 grid_3.6.1 jsonlite_1.6

[40] xtable_1.8-4 gtable_0.3.0 lifecycle_0.1.0

[43] magrittr_1.5 scales_1.0.0 stringi_1.4.3

[46] promises_1.1.0 latticeExtra_0.6-28 ellipsis_0.3.0

[49] vctrs_0.2.0 tools_3.6.1 glue_1.3.1

[52] purrr_0.3.2 crosstalk_1.0.0 fastmap_1.0.1

[55] yaml_2.2.0 colorspace_1.4-1 cluster_2.1.0

[58] knitr_1.25

**Have all data underlying the figures and results presented in the manuscript been provided?**

Reviewer #1: Yes

Reviewer #2: Yes

PLOS authors have the option to publish the peer review history of their article (what does this mean?). If published, this will include your full peer review and any attached files.

Reviewer #1: No

Reviewer #2: Yes: Dr Paul Brennan, Centre for Medical Education, School of Medicine, Cardiff University, Cardiff, Wales, CF14 4XN, United Kingdom. BrennanP@cardiff.ac.uk @brennanpcardiff

---

## [Decision Letter · Decision Letter 1]

27 Apr 2020

Dear Dr. Rutter,

We are pleased to inform you that your manuscript 'bigPint: A Bioconductor visualization package that makes big data pint-sized' has been provisionally accepted for publication in PLOS Computational Biology.

Best regards,

Aaron E. Darling

Software Editor

PLOS Computational Biology

Reviewer's Responses to Questions

**Comments to the Authors:**

Reviewer #1: The authors have addressed all my comments. I think dividing the package and manuscript by user (developer or analyst) is very helpful.

**Have all data underlying the figures and results presented in the manuscript been provided?**

Reviewer #1: Yes

PLOS authors have the option to publish the peer review history of their article (what does this mean?). If published, this will include your full peer review and any attached files.

Reviewer #1: Yes: Saskia Freytag

---

## [Editor Report · Acceptance letter]

1 Jun 2020

PCOMPBIOL-D-19-01337R1 

bigPint: A Bioconductor visualization package that makes big data pint-sized

Dear Dr Rutter,

I am pleased to inform you that your manuscript has been formally accepted for publication in PLOS Computational Biology. Your manuscript is now with our production department and you will be notified of the publication date in due course.

With kind regards,

Laura Mallard
